# Comparative Toxicological Evaluation of Tattoo Inks on Two Model Organisms

**DOI:** 10.3390/biology10121308

**Published:** 2021-12-09

**Authors:** Rosa Carotenuto, Chiara Fogliano, Mariangela Rienzi, Antonietta Siciliano, Maria Michela Salvatore, Gaetano De Tommaso, Giovanna Benvenuto, Emilia Galdiero, Marco Guida

**Affiliations:** 1Department of Biology, University of Naples Federico II, 80126 Naples, Italy; rosa.carotenuto@unina.it (R.C.); chiara.fogliano@unina.it (C.F.); m.rienzi@studenti.unina.it (M.R.); antonietta.siciliano@unina.it (A.S.); marco.guida@unina.it (M.G.); 2Department of Chemical Sciences, University of Naples Federico II, 80126 Naples, Italy; mariamichela.salvatore@unina.it (M.M.S.); gaetano.detommaso@unina.it (G.D.T.); 3Stazione Zoologica“Anton Dohrn”, 80121 Naples, Italy; giovanna.benvenuto@szn.it

**Keywords:** *Xenopus laevis*, toxicity, *Daphnia magna*, oxidative stress, gene expression, multixenobiotic resistance

## Abstract

**Simple Summary:**

Tattooing is a technique that introduces colored substances under the skin in order to color it permanently. The effects of PR170 on *Xenopus laevis* embryos and *Daphnia magna* nauplii have been studied. PR170 has nanoparticles dimensions. It modifies the survival of embryos and expression of the ATP-binding cassette in both models. Moreover, it induces deformed embryos and modifies the expression of genes involved in development and of the pro-inflammatory cytokines in *Xenopus* embryos. These effects are probably due to the oxidative stress production derived by the accumulation of PR170 and, in particular, to the presence of the azoic group in the pigment. Further studies needed to better understand the effects of commercial tattoo inks.

**Abstract:**

Tattooing is a technique that introduces colored substances under the skin in order to color it permanently. Decomposition products of tattoo pigments produce numerous damages for the skin and other organs. We studied the effects of a commercial red ink tattoo, PR170, on *Xenopus laevis* embryos and *Daphnia magna* nauplii using concentrations of 10, 20, and 40 mg/L. For *Xenopus*, we applied the FETAX protocol analyzing survival, malformations, growth, heart rate, and the expression of genes involved in the development. In *D. magna*, we evaluated the toxicity with an immobilization test. Moreover, we investigated the production of ROS, antioxidant enzymes, and the expression of the ATP-binding cassette in both models. Our results indicate that PR170 pigment has nanoparticle dimensions, modifies the survival and the ATP-binding cassette activity, and induces oxidative stress that probably produces the observed effects in both models. Deformed embryos were observed in *Xenopus*, probably due to the modification of expression of genes involved in development. The expression of pro-inflammatory cytokines was also modified in this amphibian. We think that these effects are due to the accumulation of PR170 and, in particular, to the presence of the azoic group in the chemical structure of this pigment. Further studies needed to better understand the effects of commercial tattoo inks.

## 1. Introduction

Tattooing is a technique that involves the introduction, using a solid needle, of colored substances under the skin in order to color it permanently. In recent years, we have witnessed a growth in the population of Western countries who undergo tattoos. Moreover, the areas of the body subjected to this technique are increasingly larger, covering the entire body as well. Over the years, in addition to the basic black color, numerous colors have been introduced (mainly white, red, green, yellow, and blue). The mixture that is introduced into the dermis, in addition to containing the pigments that give the color, also contains other substances such as vehicles (water, glycerin, and other alcoholic derivatives) and additives (surfactants, polycyclic aromatic hydrocarbons, nanoparticles, and polymers) [1,2]. The latest generation of tattoo inks mainly contain organic pigments (polycyclic compounds with or without the azo group) and metals such as nickel, cobalt, chromium, or lead [3,4,5,6], such as chromophores, unspecified additives, and contaminants of various types [2,7]. The composition of these elements varies according to the manufacturer and the color [2,7]. Colored tattoo pigments such as PR.22 (red azo pigment) can be decomposed by solar radiation [8,9,10] or by laser light [10,11], producing numerous decomposition products such as 2-methyl-5-nitro-aniline, which was shown to cause liver dysfunction [10,12], 4-nitro-toluene, which was shown to be genotoxic in human lymphocyte [10,13], 1,4-dichlorobenzene, which induced kidney tumors in male rats and liver tumors in male and female mice [10,14], or 2,5-dichloroaniline, which was nephrotoxic in rats [10,15]. Part of the injected dyes leave the skin through the wound left by the needles and with sweating, another fraction remains in the dermis as solid pigment particles, and a third fraction of ink is removed from the skin via the lymphatic system or blood vessels. Furthermore, some of these substances are removed from the skin by various migrating immune cells such as dendritic cells, for example, Langerhans cells in the epidermis [6]. As a result, tattoo pigments can be found at least in the lymph nodes located next to the tattoo but can also reach the organs of excretion, such as the liver or kidneys [6,16,17]. Pigment particles of tattoo suspensions have already been found in Kupffer’s cells [6,18]. For these reasons, the various dangerous substances present in tattoo inks and in their decomposition could pose a risk to health not only for the skin but also for other organs [5,6]. Consequently, people who undergo extensive tattooing risk chronic exposure to the elements contained in the inks, also considering the possibility that the tattoos need to be “refreshed” over time. It is estimated that significant amounts of ink are deposited in the dermis, ranging from about 0.60 to 9.42 mg/cm^2^ [2,8]. In humans, mainly bacterial infections and inflammatory responses are associated with tattoos, but tattoo inks have been shown to cause deleterious effects in in vitro and in vivo tests [2,16,19]. Indeed, they can be associated with cytotoxicity, oxidative stress, and p53 activation, which occurs mainly when using red and yellow inks containing azo pigments [2,20,21,22]. Arl et al. [2] investigated the composition and possible toxicological effects of four commercial tattoo inks (blue, green, red, and black) on *Daphnia magna* and HaCaT cells; they confirmed the presence of organic pigments and nanoparticles in the mixture. They concluded that red ink presented the highest toxicity, probably due to the presence of azo groups in the pigments [2]. In this context, the use of model organisms for toxicological and environmental studies would be of great help to test the toxicity of commercial tattoo inks.

The aim of our work is to understand the effects that the accumulation of a commercial red tattoo ink containing the azo dye naphthol red (also known as pigment red 170 and called PR170 from this point forward) has on two accredited models for toxicological and environmental studies: *Xenopus laevis* embryos and *Daphnia magna* nauplii. In *X. laevis*, there are no data regarding any effects produced by tattoo inks. Red inks are considered among the most dangerous between the tattoo inks [2]. We used concentrations of ink considering the quantities used in the tattoo technique [6]. For *Xenopus*, we applied the FETAX protocol and analyzed some endpoints as survival, malformations, growth, and heart rate [23,24,25]. Moreover, we investigated uptake, reactive oxygen species (ROS) and antioxidant enzymes, superoxide dismutase (SOD) and catalase (CAT) production, and the expression of some genes involved in the embryos development and of the cytokine-mediated immunological response. We also evaluated acute toxicity, uptake, ROS, SOD, and CAT production of *D. magna* nauplii exposed to PR170. The transcription of genes encoding the ATP-binding cassette was also determined in both models [26,27,28].

## 2. Materials and Methods

### 2.1. Product

The red sterile liquid tattoo ink is produced in USA and purchased in Italy, and according to the label, the ink is composed of water, CI-12475 pigment, corresponding to PR170, and witch hazel water. The manufacturers decline all responsibility for any allergic responses.

### 2.2. PR170 Characterization

The effective ink particles diameters and their size distributions were measured by transmission electron microscopy (TEM). Formvar-coated 200 mesh copper grids were used, and the excess of liquid was gently blotted using filter paper. Dried grids were directly inserted into a Jeol-JEM1220 (Jeol, Akishima, Tokyo) transmission electron microscope operating at 100 kV, and images were collected at a magnification of 50,000 using a dedicated CCD. PR170 particles were measured on the longest side, and the median, mean, and standard deviation were calculated.

### 2.3. Dynamic Light Scattering (DLS)

Dynamic light scattering (DLS) analysis was employed to determine the size of PR170 as well as to evaluate the possible aggregates formation. The sample was prepared at a concentration of 20 mg/L.

DLS measurements were performed using a home-made instrument, composed of a Photocor compact goniometer, an SMD 6000 Laser Quantum 50 mW light source (Quantum Laser Ltd., Stockport, UK) operating at 532.5 nm, a photomultiplier (PMT-120-OP/B), and a correlator (Flex02-01D) from Correlator.com (http://correlator.com/cgi-sys/, accessed on 6 November 2021) [29,30,31]. All measurements were performed at 25 °C, with the temperature controlled through a thermostatic bath, and at fixed angle *θ* = 90°.

### 2.4. Quantitative Analysis of PR170

According to the label, the tattoo ink contains PR170, water, and eventual volatile compounds not specified. Thus, the ink was completely dried with a stream of N_2_ in order to obtain a powder constituted only by the pigment. A stock solution (20 mg/L) was prepared by dissolving the required amount of pigment in double distilled water. The standard stock solution was stirred to dissolve the pigment and stored at 4 °C. A number of calibration solutions were prepared in double distilled water at a concentration range of about 2–20 mg/L and analyzed by a double beam UV–Vis spectrophotometer (Varian Cary 50, Palo Alto, CA, USA). The calibration plot was constructed by reporting (at each calibration level) the absorbance determined at the wavelength of 580 nm as a function of the known concentration of the analyte (mg/L). Visual inspection of the calibration plot showed a linear dependence of the signal from the concentration. In fact, calibration data could be excellently interpolated with a least square regression line (y=a+bx), in which *a* represents the intercept and *b* the slope (L/mg) of the least square line through the calibration points. This can be deduced from the determination coefficient (R^2^), which is very close to one (0.999). From the calibration plots, the intercept and slope were obtained together with the corresponding standard deviations ($s_{a}$ and $s_{b}$), calculated from the calibration data according to standard statistical procedures [32]. The standard deviation of the intercept, $s_{a}$, was then used to calculate the method detection limit (LOD) according to the formula: $\mathrm{LOD}=3.3\times s_{a}/b$; and the quantification limit (LOQ) according to the formula: $\mathrm{LOQ}=10\times s_{a}/b\;$ [33,34]. LOD and LOQ for PR170 were, respectively, 0.08 and 0.26 mg/L.

Concerning the solubility of PR170, it was determined starting from the preparation of a saturated solution of the pigment. It was continuously stirred in a thermostatic bath at 25 °C until the equilibrium was obtained. The concentration, measured by UV–Vis analysis, was 20.0 ± 0.8 mg/L.

Test solutions were prepared by dissolving the required volume of the ink in double distilled water. The concentration of PR170 in each solution was determined by UV–Vis analysis. For the uptake determination, the quantification of each solution was carried out in four replicates, with all data being calculated and the average values taken to represent the result.

### 2.5. Animals

#### 2.5.1. *Xenopus laevis*

*X. laevis* embryo is a very useful model for toxicological and environmental studies [23,35], but also because the data can be transferred to higher vertebrates, including humans [36,37,38]. The experiments can be performed in a short time with a large number of individuals that speed up the studies with small expenses. Moreover, *X. laevis* development takes place outside the mother’s body, allowing an easy screening of embryonic morphogenesis. The embryos are transparent, facilitating the detection of tissue and organ defects [6,23].

Adult *X. laevis* were obtained from Nasco (Fort Atkinsons, WI, USA). They were kept and used at the Department of Biology of the University of Naples, Federico II, according to the guidelines and policies dictated by the University Animal Welfare Office in agreement with international rules and in strict accordance with the recommendations in the Guide for the Care and Use of Laboratory Animals of the National Institutes of Health of the Italian Ministry of Health. The protocol was approved by the Committee on the Ethics of Animal Experiments (Centro Servizi Veterinari) of the University of Naples Federico II (Permit Number: 2014/0017970). All procedures were performed according to Italian ministerial authorization (DL 116/92) and European regulations on the protection of animals employed for experimental and other scientific purposes. All trials were adopted to minimize suffering. To obtain eggs, *X. laevis* females were injected in the dorsal lymphatic sac with 500 units of Gonase (AMSA) in amphibian Ringer solution (111 mM NaCl, 1.3 mM CaCl_2_, 2 mM KCl, 0.8 mM MgSO_4_, in 25 mM Hepes, pH 7.8). Fertilized eggs and embryos were obtained by standard insemination methods [39] and staged according to Nieuwkoop and Faber [40].

#### 2.5.2. *Daphnia magna*

Due to their unique advantages, *D. magna* is routinely used as a model organism to determine the toxicity of chemicals for risk assessment. Ephippia of *D. magna* were originally provided by ECOTOX (Milan, Italy), and a single cloned population was cultured in our lab in ISO medium [41] at a temperature of 20 °C ± 1 °C. A photoperiod of 16:8 h light–dark cycle was selected. The sensitivity of the *D. magna* clone was checked every 4 months by exposure to potassium dichromate [41]. The high-quality health status of the culture was evidenced by the low mortality rate (≤2% per week) and a high reproduction rate (about 10 neonates per day per individual). Daphnids were fed daily with microalgae *Raphidocelis subcapitata* as previously reported [42].

### 2.6. PR170 Exposure

Both models, *Daphnia magna* nauplii and *Xenopus laevis* embryos, were exposed to a concentration of 10, 20, or 40 mg/L of PR170.

#### 2.6.1. *Xenopus laevis*

We utilized modified FETAX protocol as in Tussellino et al. [39]. Embryos were collected and dejellied with β-mercaptoethanol 0.3% in Ringer solution (pH 9.0). Normally cleaved embryos at stage 4/8 cells were selected for testing and placed in 10.0 cm glass Petri dishes containing 50 mL of control or test solution. All embryos were harvested until stage 45 [35]. For each female, the plates were triplicated. All the Petri dishes were incubated in a thermostatic chamber at 21–22 °C until the end of the test, and each day, the test solutions were renewed and the dead embryos removed. For each experimental group, the number of dead larvae was recorded, and survivors were anaesthetized with MS-222 at 100 mg/L and evaluated for single malformations by examining each specimen under a stereo microscope (Leica, EZ4HD). At stage 40/41, *X. laevis* embryos opened their mouth, and ingestion became the main route of PR170 intake. Since, in the following days, the grazing behavior of the larvae became very active, the number of ingested particles increased.

#### 2.6.2. *Daphnia magna*

An acute (48 h) toxicity test was performed according to the ISO test guideline [41] (*Daphnia* sp. immobilization test), to determine the lethal concentrations of PR170 to *D. magna.* The daphnids were not fed during the test. Groups of 5 neonates (third brood, <24 h old) in 10 mL ISO medium were exposed to the three concentrations of PR170 (10, 20, 40 mg/L) for 24/48 h (*n* = 3 test groups per concentration) (ISO, 2012). After 24 and 48 h, immobilization status and abnormal appearance of *Daphnia* was checked under stereomicroscope (Leica, EZ4HD). Daphnids were considered immobile if they would not swim within 15 s after gentle agitation.

### 2.7. Xenopus laevis Phenotype Analysis and Histology

The phenotypes of embryos were scored when they had reached stage 45 at circa five days from the beginning of the treatment. We evaluated the survival, malformations, length, and cardiac frequency of the embryos as in Carotenuto et al. [43]. The embryos survival and phenotype were checked daily. All samples were photographed with a Leica MZ16F UV stereomicroscope, equipped with a Leica DFC 300Fx camera and IM50 Image Manager Software.

For histology, the embryos were performed with standard histological protocols for optical microscopy as in Carotenuto et al. [35]. Briefly, after euthanasia with 2 mg/L of tricaine (MS-222, Sigma–Aldrich, Denmark) the samples were fixed in Bouin solution for 24 h, dehydrated, and mixed with paraffin. Hematoxylin–eosin staining was performed on Sections of 5 μm [23]. At least 10 slides of 10 different embryos from each treatment were examined, and the morphology of the eyes, gut, and gills were investigated. The images were acquired with a Zeiss Axiocam Microscope Camera Applied to a Zeiss Axioscope microscope (Zeiss, Jena, Germany).

### 2.8. Real-Time PCR Analysis

Target gene mRNA was quantified with quantitative real-time PCR. All analyses were carried out using the Applied Biosystem 7500 Real-Time PCR System and the Power SYBR^®^ Green PCR Master Mix (Life Technologies, MA, USA) in 20 µL total reaction volume following procedures recommended by the manufacturer. RNA was extracted with the Direct-zol RNA Mini Prep kit (ZymoResearch, Irvine, CA, USA) following the manufacturer’s instructions and used for cDNA synthesis using the SuperScript^®^ VILO cDNA synthesis kits (Life Technologies) and 2 µg of total RNA. Used primers are indicated in Table 1. The amplification thermal profile is as follows: 95 °C for 3 min (initial DNA denaturation); 40 cycles of 95 °C for 15 s; and 60 °C for 1 min. A final dissociation curve assay was performed for each reaction to confirm gene-specific amplification.

### 2.9. ROS Production and Antioxidant Activity Analysis

After each exposure, *X. laevis* embryos and *D. magna* neonates were collected to analyze ROS production and the activation of antioxidant defense including superoxide dismutase (SOD) and catalase (CAT). Each sample was homogenized with 0.1 mL of 50 mM potassium phosphate-buffered solution (PBS) (pH 7.4) using a sterile pestle. Homogenates were centrifuged for 20 min at 15,000 rpm (4 °C). The protein concentration of each sample was measured in three replicates using a spectrophotometer (Hach-Lange DR 5000, Düsseldorf, Germany) according to the Bradford method [44]. ROS content and activity of SOD and CAT were measured using Sigma-Aldrich kits according to manufacturer’s similarly to [42].

ROS content was detected using flowing fluorescent dye 2′,7′dichlorodihydrofluorescein diacetate (H_2_DCFDA) and quantified by the ability of radicals to oxidize the H2DCFDA to a fluorescent product that was measured spectrophotometrically, with an excitation wavelength of 350 nm and an emission wavelength of 600 nm. CAT activity was quantified by decrease in absorbance at 240 nm due to H_2_O_2_ consumption. SOD activity was detected using the WST-1 [2-(4-iodophenyl)- 3-(4-nitrophenyl)-5-(2,4-disulfophenyl)-2Htetrazolium, monosodium salt] that produces a water-soluble formazan dye upon reduction with a superoxide anion and measuring the decrease in the color development at 440 nm.

### 2.10. Statistical Analysis

Statistical analysis was carried out by using a GraphPad Prism 8 software (GraphPad, San Diego, CA, USA). Results are given as mean ± standard deviation. Differences between control and treatment groups were determined through an unpaired two-tailed Student’s *t* test.

The percent of embryos death has been verified by Chi-square test, and the death distributions were assessed in terms of significance using a Mantel–Cox test. The frequency of malformations was estimated on survival embryos. All data were examined by Chi-square test using Fisher’s exact test. Real-time PCR statistical analysis was performed with two-way ANOVA with Bonferroni’s correction. Differences were considered significant when *p* < 0.05.

## 3. Results

### 3.1. PR170 Had Nanoparticles Dimensions

Electron microscopy showed that pigment contained in our red tattoo ink had nanoparticles dimensions. These NPs were electron-dense (Figure 1A), had a polygonal shape (Figure 1A) and an average length of 110 nm, but their dimensions do not exceed 250 nm (Figure 1B). They probably form agglomerates (Figure 1A, arrow).

The hydrodynamic dimension of PR170 was determined from the diffusion coefficient of particles present in the solution. In the approximation of spherical objects, continuous medium and infinite dilution, the diffusion coefficient can be easily related to the hydrodynamic radius R_h_ through the Stokes–Einstein equation:$$R_{h}=\frac{k_{B}T}{6\pi \eta D},$$
where k_B_ is the Boltzmann constant, T is the absolute temperature, and η is the solvent viscosity. For non-spherical particles, and R_h_ represents the radius of a spherical aggregate with the same measured diffusion coefficient.

The DLS measurements show a polydisperse system. As can be noticed in Figure 2, there are three populations of about 113, 35, and 9 nm (see black line). This representation enhances large aggregates, which efficiently scatter the light. The DLS is, in principle, more sensitive to large objects than to smaller ones, with the intensity proportional to the sixth power of radius; hence, the red line in Figure 2 was calculated by converting the intensity-weighted profiles into a number-weighted profile, obtaining an indication of the concentration of the different species in the sample. This second representation shows a most abundant population of about 9 nm, and the other of about 30 nm. Therefore, this means that the system tends to aggregate in solution, and this result is in agreement with the TEM images.

### 3.2. PR170 Changes the Mortality Rate of Xenopus laevis Embryos and Daphnia magna Nauplii

In *X. laevis*, PR170 uptake rates increased as concentrations increased. A significant reduction in PR170 in solution (87.9%) was observed when *X. laevis* was exposed to the initial concentration of 20 mg/L. A smaller reduction in the uptake rate (55.4%) was observed at 10 mg/L (Table 2). 

The *Xenopus* embryos treated with PR170 (*n* = 540, *p* < 0.01) significantly change the percentage of death, which reaches 47.41% at a concentration of 20 mg/L, therefore, with an increase in percentage death of 18% regarding the controls. The percentage of death is 34.81% and 32.59% at 10 mg/L and 40 mg/L, respectively (Figure 3A and Table 3), which is similar to the range of control embryo mortality (29.63%) (Figure 3A and Table 3). 

Under PR170 exposure, *D. magna* retained the dye in the gut both at a concentration of 10 mg/L and 20 mg/L of PR170 (Figure 4B). It was relevant that the uptake in the gut increased as the exposure concentration decreased as supported by the chemical measurements of PR10 in exposure media. Indeed, we observed that PR170 is reduced by 18.2% to 10 mg/L and by 23.2% to 20 mg/L of the initial concentration.

In the acute toxicity test, the control group showed no immobility. Concentration–response relationships were obtained, in fact all *D. magna* nauplii were found dead when exposed to 40 mg/L of PR170. The animals showed lower mortality when they were treated with 20 mg/L of PR170 with 76% (±9.5% dev. std.; *n* = 3) of immobility or 10 mg/L of PR170, with only 28% (±14% dev.std.; *n* = 3) of negative effects (Figure 4A).

Our data showed that, in both models, the 20 mg/L concentration of PR170 produces the major percent of mortality.

### 3.3. PR170 Does Not Change the Growth Rate but Causes Anomalies and Tachycardia in Xenopus laevis Embryos

Treated embryos showed unmodified length compared to the control (*n* = 120, *p* ˃ 0.05) (Figure 3B), but they presented abnormalities in a small percentage (Table 3 and Figure 5 and Figure 6). These malformations are visible at all concentrations, but with higher frequency in embryos treated with 20 mg/L of PR170 where exceeded the 25%, twice as much as the control (about 12%) (Table 3). The anomalies observed mainly concerned the presence of very extensive edema of the abdomen and heart areas, 7, 10, and 9% at 10, 20, and 40 mg/L, respectively (Figure 5D–F, asterisks). The distribution of the pigment was modified, with 7, 13, and 11% at 10, 20, and 40 mg/L, respectively (Figure 5D,G, arrowhead), and in some cases, the embryos were completely pigmented (Figure 5G). The PR170 pigment completely invaded the intestine (Figure 5C (arrow) and Figure 6B’, arrowhead) adhering tightly to the walls (Figure 6B”, double arrows), while it was not visible in the gills (Figure 6C, asterisk) or other organs. In some cases, the intestine protruded from the abdomen (Figure 5, double arrows); less frequently, we observed folded tails (Figure 4B,D, white line), misshapen eyes (Figure 5D and Figure 6A’, empty arrowhead), or eyes different in size from the controls (Figure 5E, empty arrowhead). Treated embryos (*n* = 144, *p* < 0.0001) showed modification of the heart rate; they are, in fact, tachycardic compared to the control, in particular, embryos treated with 40 mg/L of PR170 (Figure 3C).

### 3.4. PR170 Modifies the Expression of Genes Involved in Early Embryonic Development and of Pro-Inflammatory Cytokines of X. laevis

We explored the possible alterations of gene expression caused by PR170 in *X. laevis*. Real-time PCR, conducted on five-day-old embryos, showed a general modification of the expression of genes involved in the early development and of pro-inflammatory cytokines in treated embryos (Figure 7A).

We verify the expression of genes involved in early development as *bmp4* and *fgf8* that contribute to the formation of embryonic axes, *sox9* and *egr2* that specify the neural crests and their movement, and *pax6* and *rax1* involved in encephalon and eyes morphogenesis.

At 10 mg/L of PR170, *bmp4* (*p* < 0.01) was overexpressed, *fgf8* and *egr2* were slightly overexpressed, while *sox9* (*p* < 0.001), *pax6* (*p* < 0.0001), and *rax1* were downregulated. We observed a downregulation of all studied genes at 20 mg/L of PR170 (*p* < 0.0001). At 40 mg/L of PR170, excluding *bmp4* expression, which was comparable to the control (*p* ˃ 0.05), all genes were downregulated; *fgf8* and *egr2* with *p* < 0.01; and *sox9*, *pax6*, and *rax1* with *p* < 0.0001. Therefore, PR170, at all concentrations, interferes with expression of these genes that are very important during early development of *Xenopus*.

To verify the immunological response of embryos after PR170 treatment, we analyzed the expression of *tnfα* and *il1b*, the cytokines involved in the activation of NF-kappa-B transcription factor, and *p65*, involved in NF-κB heterodimer formation, as well as nuclear translocation and activation (Figure 7B). We observed that *tnfα* was downregulated at 10 (*p* < 0.001) and 20 (*p* < 0.01) mg/L of PR170, while *il1b* was upregulated at 20 and 40 mg/L (*p* < 0.0001) and slightly downregulated at 10 mg/L; *p65* was always downregulated, in particular, at 20 and 40 mg/L (*p* < 0.001) of PR170. The modifications of expression of these genes indicate that PR170 always induce the activation of immune system in treated embryos.

### 3.5. PR170 Modify the Activity of ATP-binding Cassette in Daphnia magna and Xenopus laevis

In *Daphnia magna*, we analyze the expression of *abcb1* (ATP-binding cassette subfamily B member 1) and *abcc1/3*, *abcc4*, and *Abcc5* (ATP-binding cassette subfamily C) genes according to Campos et al. [45]. Our data showed that, in treated *Daphnia*, *abcc1/3* was slightly overexpressed at 10 mg/L, and *abcc4* was always upregulated. In contrast, *abcb1* and *Abcc5* were always downregulated, in particular, at 20 mg/L (Figure 7C). These patterns point out that exposure treatments affect significantly (*p* < 0.05) the transcription of the studied genes only at lower concentrations.

Moreover, in *X. laevis*, we analyzed the expression of the *Abcb1* (Figure 7B), a member of the MDR/TAP subfamily (superfamily of ATP-binding cassette transporters) involved in multidrug resistance (provided by RefSeq, February 2017). We showed that *abcb1* was expressed less and less as the concentration increased (*p* < 0.05). This trend indicates that PR170 interferes with activity of ATP-binding cassette by inhibiting it.

### 3.6. PR170 Induces Oxidative Stress

The increase in ROS and the sensitivity in enzymatic activity varied due to the function of each enzyme and depended on the test species, despite the use of the same chemicals.

Figure 8A shows ROS accumulation for *D. magna*, as can be observed in animals exposed to 10 mg/L and 20 mg/L of PR170; these show a significant increase of about 30 and 40%, respectively, compared to controls after 30 min. Regarding the antioxidant enzymes, the CAT activity (Figure 8B) shows a significant antioxidant activity at concentration of 10 mg/L than at 20 mg/L.

In *X. laevis*, the percentage increases in ROS were 65, 81, and 84% compared to the control, corresponding to 10, 20, and 40 mg/L, respectively (Figure 8A). The CAT-activity increments showed a U-shaped dose-dependent activity (Figure 8B); SOD activity increased from 10 to 20 mg/L as compared to the control and decreased at 40 mg/L, showing an inverted U-shaped curve (Figure 8C).

## 4. Discussion

Modern tattoo inks mainly contain organic pigments, metals, unspecified additives, and contaminants of various types [2,6,7] mixed in various way. Use of model organisms to study their possible toxic effects is certainly of great help.

Our study was inserted in this context; in fact, we have studied the toxic potential of a commercial red ink tattoo on two models, *X. laevis* and *D. magna*, using different approaches. TEM and DLS data showed that it was made of nanoparticles, according to Hogsberget al. [1], and was an aggregate in part; they are polygonal shaped. Arl et al. [2] showed the presence of nanoparticles and aggregates in some tattoo inks. Then, we analyzed the uptake of these particles, and we have shown that PR170 is strongly absorbed by *Xenopus*, while in *Daphnia*, the absorption percentage is much lower; we must consider that, while the assumption of PR170 in amphibians can certainly occur through ingestion and the respiratory system but also through the skin, on the contrary, crustaceans have a chitinous coating that probably partially blocks their entry. However, our images show a large presence of pigment in the intestine in both cases; Arl et al. [2] showed the presence of the ink in gut and adherent to exoskeleton of *D. magna*.

A FETAX assay showed that PR170 causes a high level of death of treated *Xenopus* embryos only at a concentration of 20 mg/L; we think that, probably, a 10 mg/L concentration is insufficient to cause death, and that at 40 mg/L, the particles partially precipitate and then fail to reach the concentration in the solution that causes death [39]. PR170 does not modify the growth of embryos but induces, in a small percentage, very serious malformations of the surviving embryos. The most frequent malformations concern very extensive edema that compromised the internal organs such as the heart by tachycardic and pigment distribution. We also observed problems in the eyes, which appeared deformed or of different size, in the digestive system, which, in some cases, extruded from the body, and in tails [39,42,46]. The observed malformations we induced to verify the expression of a panel of genes involved in early development, and we have found that, in treated embryos, if the expression of these genes is modified, they are, in general, downregulated [23]. In particular, at 20 mg/L, the concentration showing the highest percentage of malformations, all genes were downregulated. We must consider that *bmp4* and *fgf8* are involved in axes formation [47,48,49], and *sox9* and *egr2* are involved in the determination and positioning of the neural crests, the fourth embryonic sheet; this could be at the basis of the abnormal production of melanocytes and their distribution and cardiac problems [50,51]. Finally, *pax6* and *rax1* are involved in the correct morphogenesis of the anterior encephalon and eyes [52,53]. Oxidative stress could be at the basis of the modified gene expression [51,54]. Treated *Xenopus* embryos showed high production of ROS and of antioxidant enzymes SOD and CAT, indicating that these embryos were subjected to strong oxidative stress. Arl et al. [2] showed a high production of ROS and antioxidant enzymes in *D. magna* and HaCaT cells treated with tattoo inks. The dose-dependent U-shape observed for some parameters, such as the oxidative enzyme CAT, could be explained considering an hormetic effect between the concentrations of 10 and 20 mg/L, while at 40 mg/L, probably, the formation of aggregates reduces the quantity of PR170 that penetrate, and then the effects return similar to those observed at 10 mg/L [55]. Moreover, oxidative stress is known to modulate the generation of inflammatory cytokines through activation of NF-κB [56]. Our treated *Xenopus* embryos showed downregulation of *tnfα* and *p65* and a high expression of *il1b* [57]. In mice neutrophils, prolonged pharmacologic inhibition of NF-kB augments IL-1b secretion [58]. These modifications indicate that the PR170 induces an immune response in treated embryos. Finally, we investigated whether the treated *Xenopus* embryos were able to activate a defense mechanism from xenobiotic compounds by studying the expression of *abcb1* involved in multidrug resistance. We showed that *abcb1* downregulated at 20 mg/L and 40 mg/L; these results indicate that the activity of *abcb1* decreases as concentration increases, contributing to the inflammatory state not being able to expel toxic substances contained in the red ink [35].

As regards the toxicity of PR170 to *D. magna*, this study provides major information to evaluate the risks on aquatic ecosystems. Our results did not differ from previous studies. The results from an acute toxicity test had shown that PR170 was toxic already at concentrations of 10 mg/L, confirmed by the uptake that was greater at the lower concentrations tested. The use of oxidative stress biomarkers has become a promising tool to evaluate the toxic effects of environmental pollutants as early indicators in ecotoxicology, so we observed a disruption of the balance of biological oxidant-to-antioxidant system. The strongest oxidative stress and the antioxidant activity were induced when exposed at 10 mg/L indicating that PR170 generates oxidative stress, leads to the induction of oxidative damages on *D. magna* and negatively influences biological homeostasis in the organism. The expression levels of five analyzed genes related to the cellular multixenobiotic resistance (MXR) system, which have been proposed as a first line of defense against environmental toxicants, confirm the activities inhibition leading to decreases in tolerance against this toxicant. In fact, the modification of *abcb1*, *abcc1/3*, *abcc4*, and *Abcc5* genes’ expression determined after exposure to PR170 could suggest it is probably a substrate of MXR transporters in *D. magna*, confirming the existence of a regulation system to limit excessive accumulation of toxicants.

## 5. Conclusions

Our results in complex indicate that PR170 initiated strong oxidative stress in both tested organisms, as manifested by the high increase in ROS that were, probably, not neutralized by the antioxidant enzymes. This implies that the antioxidant defense system was inefficient in protecting the organisms against the oxidative stress triggered by PR170, leading to oxidative damages. This situation has produced the effects we have shown on the inflammatory state and the attempt to clear the toxic elements carried out by *Xenopus* and *Daphnia*, but also influenced the expression of genes involved in embryonic development leading to the production of deformed embryos, albeit in a small percentage. We think that these effects are due mainly to the size and accumulation of PR170 and to the presence of the azoic component in the tattoo ink we tested. In light of our data, we believe that further studies needed to better understand the effects produced by tattoo inks and efficient control over the compositions of tattoo inks available on the market.

## Figures and Tables

**Figure 1 biology-10-01308-f001:**
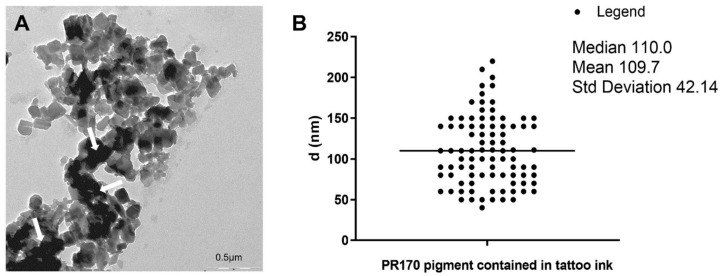
PR170 characterization. (**A**) TEM micrography of PR170 highlighted their polygonal shape and the formation of agglomerates (arrow). (**B**) Distribution of diameters of PR170 contained in tattoo ink. Average diameter is 110 nm.

**Figure 2 biology-10-01308-f002:**
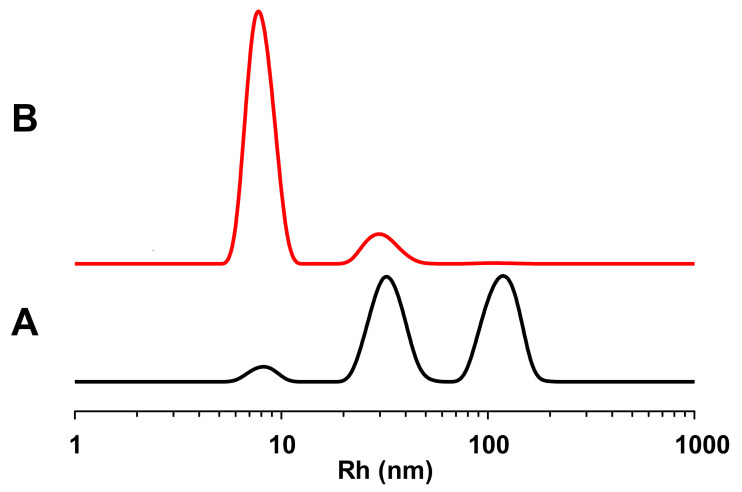
Graphical representations of DLS measurements of PR170. (**A**): hydrodynamic radii distribution (9 ± 1, 35 ± 9 and 113 ± 9); (**B**): normalized hydrodynamic radii distribution (9 ± 1 and 30 ± 1).

**Figure 3 biology-10-01308-f003:**
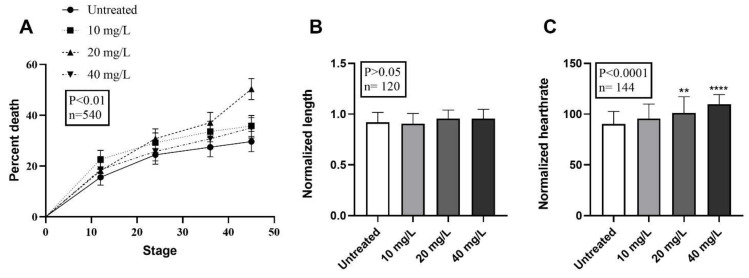
Mortality, growth retardation, and heartrate evaluations after PR170 pigment treatment in *X. laevis* embryos. (**A**) Mortality percentage distributions of untreated and treated embryos display a peak at 20 mg/L and values not so far from the control for 10 mg/L and 40 mg/L. (**B**) Analysis revealed unmodified length of treated embryos if compared to the wild type (*p* > 0.05). (**C**) Treated embryos showed a growing tachycardia with maximum rate at 40 mg/L. ** *p* < 0.01, **** *p* < 0.0001.

**Figure 4 biology-10-01308-f004:**
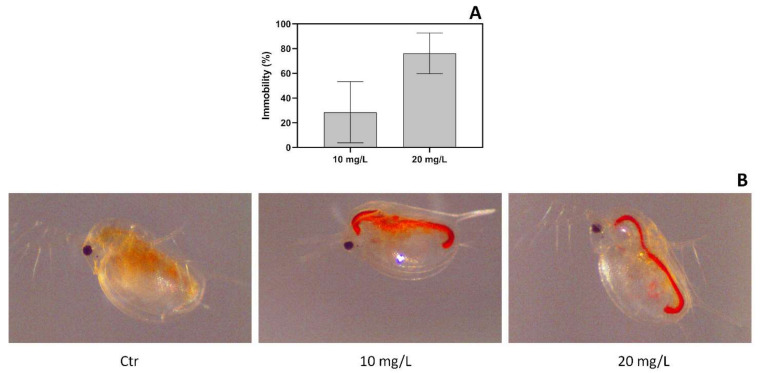
*Daphnia magna* exposure. Immobility of *Daphnia magna* after 48 h of exposure (**A**); *D. magna* exposed to PR170 samples at different concentration (**B**): Control (Ctr), 10 mg/L and 20 mg/L. In the control (Ctr) is visible a fasting gut, in the 10 mg/L and 20 mg/L exposures PR170 accumulation is visible inside the digestive system.

**Figure 5 biology-10-01308-f005:**
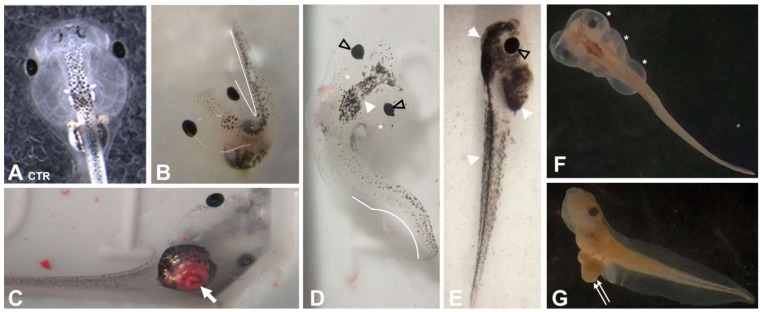
Effects of treatment with PR170 on *X. laevis* embryos development. (**A**) Control. (**B**) Presence of embryos with fold tail (white line). (**C**) Red pigment invasion of the intestine (arrow). (**D**) Presence of head edema (asterisk) accompanied by fold tail (white line), misshapen eyes (empty arrowhead), and increase in pigment (white arrowhead). (**E**) Excessive pigmentation (white arrowhead) and presence of bigger eyes (empty arrowhead). (**F**) Widespread edema of head and abdomen (asterisks). (**G**) Intestine protrusion from the abdomen (double arrows).

**Figure 6 biology-10-01308-f006:**
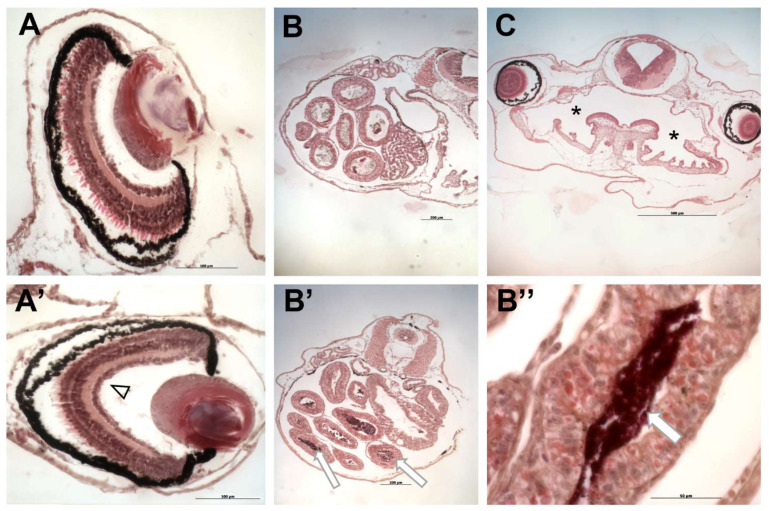
Histological analysis of *X. laevis* embryos treated with PR170 pigment. (**A**) Control eye. (**A’**) Embryos treated with PR170 pigment in some cases showed eyes with elongated shape (empty arrowhead). (**B**) Control intestine. (**B’**) Intestine of treated embryos showed the presence of pigment in the intestinal loops (arrows) with tight adhesion to the walls (**B’’**, arrow). (**C**) Gills of treated embryos were not invaded by pigment (asterisk). Magnification: (**B**,**B’**,**C**): 5×; (**A**,**A’**): 20×; (**B’’**): 40×.

**Figure 7 biology-10-01308-f007:**
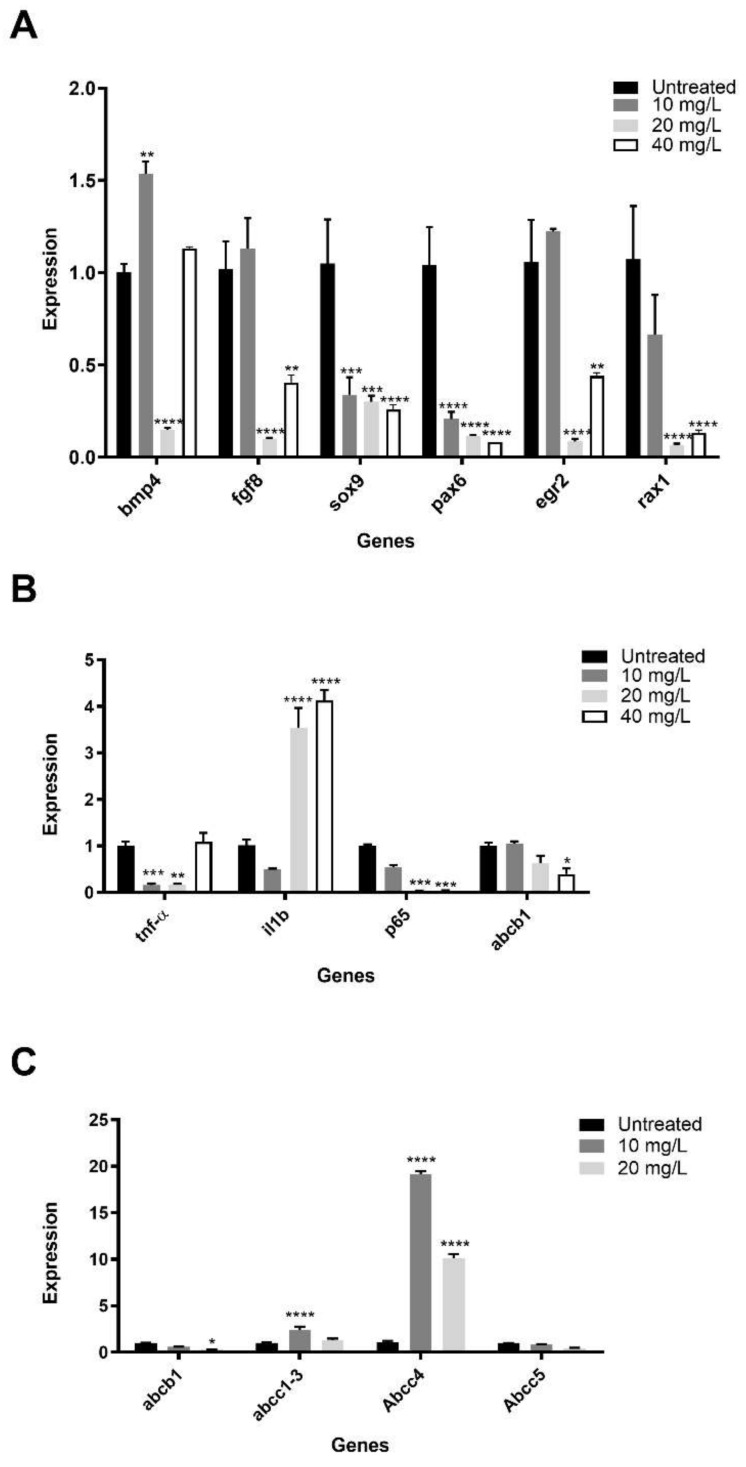
Gene expression in *X. laevis* and *D. magna*. Expression of genes involved in early embryonic development (**A**) and inflammation (**B**) of *Xenopus laevis*. ATP binding cassette mechanism expression was showed in *Daphnia magna* (**C**) and *Xenopus laevis* (**B**). Data are presented as mean with SD. Statistical significance was determined using *t*-tests Hold Siddak correction for multiple comparison. * *p* < 0.05, ** *p* < 0.01, *** *p* < 0.001, **** *p* < 0.0001.

**Figure 8 biology-10-01308-f008:**
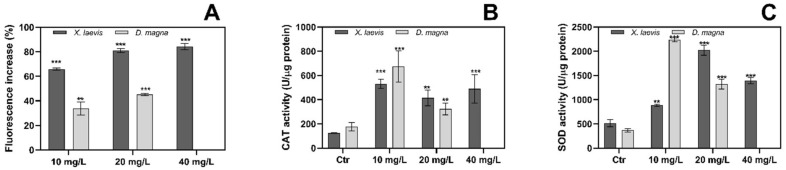
Oxidative stress in *X. laevis* and *D. magna*. Effects of PR170 on (**A**) ROS production expressed as fluorescence intensity, (**B**) CAT activity, and (**C**) SOD activity. The level of significance was set at ** *p* < 0.01, *** *p* < 0.001.

**Table 1 biology-10-01308-t001:** Primers.

Gene Name	Oligo Forward Sequence	Oligo Reverse Sequence
bmp4—bone morphogenetic protein 4	CCTCAGCAGCATTCCAGAGAA	TCCGGTGGAAACCCTCATCC
egr2—early growth response 2	AGTAAGACCCCAGTCCACGA	GCAGTAATCGCAGGCAAAGG
fgf8—fibroblast growth factor 8	CGTTTGGAAGCAGAGTTCGC	GTTGCCTTGTCTTCGACCCT
odc1—ornithine decarboxylase	GTGGCAAGGAATCACCCGAA	TCAAAGACACATCGTGCATC
pax6—paired box protein pax-6	CAGAACATCTTTTACCCAGGA	GAATGTGGCTGGGTGTGTTA
rax1—retinal homeobox protein Rx1	GGAAAGACCTCAAGCGAGTG	ATACCTGCACCCTGACCTCG
sox9—sex determining region Y-box 9	ACGGCGCAGAAAGTCTGTTA	GACATCTGTCTTGGGGGTGG
ATPbc—ATP binding cassette, subfamily B member 1	GGCTGTTGCTGAAGAGGTTC	ACCATACCAAAAGGCGAGTG
TNFa—tumor necrosis factor alfa	CAAGCAATGAAAGGGGAAAA	TGCAGTCAGGACCTGTGAAG
IL1B—interleukin 1 beta	TGTGCAGATAACCCATGGAA	TGCAGAGCAACAGAAGATGG
p65—Nf-kB transcription factor family	TGGCTATTGTCTTCCGAACC	ATATGGTGGGGGTCTCCTTC
β—actin daphnia	TTATGAAGGTTACGCCCTGC	GCTGTAACCGCTTCAGTCAA
abcb1- ATP Binding Cassette Subfamily B Member 1	GTATCCAGTGCGGAAGTGGC	ACAGCGTATCGCTATTGCCC
abcc1/3—ATP Binding Cassette Subfamily C Member 1	TAGCTCGCGCTCTACTGAGAA	GATCGTCGGTCTCCAGATCG
abcc4—ATP Binding Cassette Subfamily C Member 4	CCCGATCCCTTTACGTCGAT	GGTGGCGTCCTACATGAGTGT
abcc5—ATP Binding Cassette Subfamily C Member 5	CAGTCCAGTCATCGAGAACGG	TGACGCAACAGAGCTCGG

**Table 2 biology-10-01308-t002:** Concentration of pigment red 170 in test solutions of tattoo ink and uptake determination in *Xenopus laevis* and *Daphnia magna*. Data are presented as average ± standard deviation (SD).

Nominal Concentration before Treatment (mg/L)	Concentration (from UV-Vis Analysis) before Treatment (mg/L) ± SD	Concentration (from UV-Vis Analysis) after Treatment (mg/L) ± SD	Uptake (%)
*Xenopus laevis*			
10.0	9.2 ± 0.2	4.1 ± 0.4	55.4
20.0	17.3 ± 0.3	2.1 ± 0.2	87.9
*Daphnia magna*			
10.0	9.9 ± 0.4	8.1 ± 0.5	18.2
20.0	19.8 ± 0.3	15.2 ± 0.6	23.2

**Table 3 biology-10-01308-t003:** *Xenopus laevis* embryo mortality and malformations.

	Mortality				Malformations
	Utilized (*n*)	Dead (*n*)	Living (*n*)	Mortality (%)	*n* (%)
Untreated	135	40	95	29.63 ^b^	12 (12.63) ^b^
10 mg/L	135	47	88	34.81 ^b^	18 (20.45) ^b^
20 mg/L	135	64	71	47.41 ^a^	18 (25.35) ^b^
40 mg/L	135	44	91	32.59 ^b^	21 (23.08) ^b^

*n* = number embryos used; ^a^ Chi square test *p* < 0.01; ^b^ Chi square test *p* > 0.05.

## Data Availability

All data generated or analyzed during this study are included in this published article.

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
