# Peer review of "Comparative Toxicological Evaluation of Tattoo Inks on Two Model Organisms"

_biology, 2021, doi:10.3390/biology10121308_

Round 1

Reviewer 1 Report

The comparative toxicological evaluation of tattoo inks is a useful contribution to the safety assessment of these substances. Evaluation was performed according to existing standards.

Comments

-Description of the Materials and methods section is incomplete:

Methodology of RNA isolation should be added and the methodology of the used commercially available assays for ROS, CAT and SOD activity briefly described. Organs that were evaluated in the histological analysis should be listed.

-For evaluation of potential oral uptake hydrodynamic size is very important to identify agglomeration/precipitation and these data should be added.

-There is also no information provided, to which extent dissolution may take place.

-A table on the observed malformations and their frequency would be helpful. Is there organotropy?

-Dose-dependency for several parameters is U-shaped – what is the authors’ explanation for this?

-Why is downregulation of tnfa interpreted as pro-inflammatory effect? (it is usually regarded as pro-inflammatory cytokine)

Minor

  • Several sentences should be corrected regarding the wording/grammar or typing errors (l.95, l.110, l.273, l.407, l.420) and missing spaces between words added.

Author Response

Comments and Suggestions for Authors

The comparative toxicological evaluation of tattoo inks is a useful contribution to the safety assessment of these substances. Evaluation was performed according to existing standards.

Dear Sir or Madam,

we appreciate your careful reading of our manuscript and valuable suggestions. We have carefully considered the comments and revised the manuscript accordingly. All of the corrections are highlighted by red text in the revised manuscript and summarized point by point as follows:

 Comments

-Description of the Materials and methods section is incomplete:

Methodology of RNA isolation should be added and the methodology of the used commercially available assays for ROS, CAT and SOD activity briefly described.

Response: according to suggestion we have added the methodology required

-Organs that were evaluated in the histological analysis should be listed.

Response: according to suggestion we have indicated the organs evaluated in “Material and Methods”, section 2.6.

-For evaluation of potential oral uptake hydrodynamic size is very important to identify agglomeration/precipitation and these data should be added.

Response: According to the referee comment, DLS measurements were conducted and data concerning the hydrodynamic size of PR170 were added. In particular, in the new version of the manuscript a new subparagraph of materials and methods (see subparagraphs 2.3) was added and the results section was integrated with the new data (see subparagraph 3.1 and Figure 2).

-There is also no information provided, to which extent dissolution may take place.

Response: According to the referee comment, this information was added in the subparagraph 2.4

-A table on the observed malformations and their frequency would be helpful. Is there organotropy?

Response: according to suggestion we have added the percentage of total malformations in  “Table 2” and the percentage of single malformations in “Results” (section 3.3.). We did not investigate whether the PR170 was organotropy, but we observed anomalies only in some organs.

-Dose-dependency for several parameters is U-shaped – what is the authors’ explanation for this?

The dose-dependence U-shaped observed for some parameters, as the oxidative enzyme CAT, could be explained considering  an hormetic effect between the concentrations of 10 and 20 mg/L  and  that at 40 mg/L, probably, the formation of aggregates reduces the quantity of PR170 that penetrate and then the effects return  similar to the effects observed at 10 mg/L.

-Why is downregulation of tnfa interpreted as pro-inflammatory effect? (it is usually regarded as pro-inflammatory cytokine)

 Response: we have modified our sentences regarding the cytokines responses.

Minor

-Several sentences should be corrected regarding the wording/grammar or typing errors (l.95, l.110, l.273, l.407, l.420) and missing spaces between words added.

Response: according to suggestion we have modified

Reviewer 2 Report

I have reviewed the manuscript entitled "Comparative Toxicological Evaluation of Tattoo Inks on Two Model Organisms" by Carotenuto et al. for publication in Biology. The authors have investigated the effects of a commercial red ink tattoo (at concentrations of 10, 20 and 40 mg/L) on Xenopus laevis embryos and Daphnia magna nauplii. The work is well organized and the subject matter is interesting, so I recommend it for publication after minor revision.

Comments:

1) References are not in the journal format.

2) Check spacing and punctuation along the text (for example in lines 29, 114, 172 etc).

3) Lines 226-227 “Homogenates were centrifuged for 20 min at 15,000g (4 °C)”: sentence not clear, rewrite in a clearer manner.

4) Statistical Analysis: inconsistency between line 234 “Results are given as mean ± standard error” and line 274 “Data are presented as average ± standard deviation (SD)”. Is the standard deviation or standard error shown?

5) Include comparisons with literature data of similar research to better highlight the result of your study.

Author Response

I have reviewed the manuscript entitled "Comparative Toxicological Evaluation of Tattoo Inks on Two Model Organisms" by Carotenuto et al. for publication in Biology. The authors have investigated the effects of a commercial red ink tattoo (at concentrations of 10, 20 and 40 mg/L) on Xenopus laevis embryos and Daphnia magna nauplii. The work is well organized and the subject matter is interesting, so I recommend it for publication after minor revision.

Dear Sir or Madam,

we appreciate your careful reading of our manuscript and valuable suggestions. We have carefully considered the comments and revised the manuscript accordingly. All of the corrections are highlighted by red text in the revised manuscript and summarized point by point as follows:

Comments:

  • References are not in the journal format.

Response: according to suggestion we have formatted the reference

  • Check spacing and punctuation along the text (for example in lines 29, 114, 172 etc).

Response: according to suggestion we have modified

  • Lines 226-227 “Homogenates were centrifuged for 20 min at 15,000g (4 °C)”: sentence not clear, rewrite in a clearer manner.

Response: according to suggestion we have clarified

4) Statistical Analysis: inconsistency between line 234 “Results are given as mean ± standard error” and line 274 “Data are presented as average ± standard deviation (SD)”. Is the standard deviation or standard error shown?

Response: according to suggestion this inconsistency was corrected throughout the text.

  • Include comparisons with literature data of similar research to better highlight the result of your study

Response: according to suggestion we have added and compared new references

Round 2

Reviewer 1 Report

Thank you for the revision, I have no further comments.